# Extracellular Matrix Regulation in Physiology and in Brain Disease

**DOI:** 10.3390/ijms24087049

**Published:** 2023-04-11

**Authors:** Alyssa Soles, Adem Selimovic, Kaelin Sbrocco, Ferris Ghannoum, Katherine Hamel, Emmanuel Labrada Moncada, Stephen Gilliat, Marija Cvetanovic

**Affiliations:** 1Department of Neuroscience, University of Minnesota, 321 Church St SE, Minneapolis, MN 55455, USA; 2Institute for Translational Neuroscience, University of Minnesota, 2101 6th Street SE, Minneapolis, MN 55455, USA

**Keywords:** extracellular matrix, regulation, gene expression, microglia, neurons, HIF-1, perineuronal nets

## Abstract

The extracellular matrix (ECM) surrounds cells in the brain, providing structural and functional support. Emerging studies demonstrate that the ECM plays important roles during development, in the healthy adult brain, and in brain diseases. The aim of this review is to briefly discuss the physiological roles of the ECM and its contribution to the pathogenesis of brain disease, highlighting the gene expression changes, transcriptional factors involved, and a role for microglia in ECM regulation. Much of the research conducted thus far on disease states has focused on “omic” approaches that reveal differences in gene expression related to the ECM. Here, we review recent findings on alterations in the expression of ECM-associated genes in seizure, neuropathic pain, cerebellar ataxia, and age-related neurodegenerative disorders. Next, we discuss evidence implicating the transcription factor hypoxia-inducible factor 1 (HIF-1) in regulating the expression of ECM genes. HIF-1 is induced in response to hypoxia, and also targets genes involved in ECM remodeling, suggesting that hypoxia could contribute to ECM remodeling in disease conditions. We conclude by discussing the role microglia play in the regulation of the perineuronal nets (PNNs), a specialized form of ECM in the central nervous system. We show evidence that microglia can modulate PNNs in healthy and diseased brain states. Altogether, these findings suggest that ECM regulation is altered in brain disease, and highlight the role of HIF-1 and microglia in ECM remodeling.

## 1. Introduction

The extracellular matrix (ECM) is the scaffold in which cellular components of all tissues are embedded. It constitutes roughly 40% and 20% of the total brain volume of the developing and adult brains, respectively [1]. While brain-specific forms of the ECM were originally described by Camillo Golgi in 1882 [2], several recent studies are unraveling the dynamic nature of ECM’s composition and regulation, and its active role in development, adulthood, and the pathogenesis of brain diseases.

The ECM is composed of a mix of proteins and carbohydrates. Structured fibrous proteins such as elastin, laminins, and collagens form an organized scaffold (Figure 1A). A less structured, amorphous gel made of hyaluronan (HA), proteoglycans, tenascins, link proteins, and glycoproteins fills in around the scaffold. The ECM connects to cells by binding to ECM receptors, including integrins, CD44, a receptor for hyaluronan-mediated motility (RHAMM), Toll-like receptors 2 and 4 (TLR-2, TLR-4), and NCAM [3]. In addition, the ECM contains secreted molecules such as growth factors, neurotransmitters (NTs), ions, and neuromodulatory agents.

The ECM of the central nervous system (CNS) differs from the ECM of other organs by the predominance of non-fibrous components and two specialized forms of the ECM. The first specialized form is perineuronal nets (PNNs), which mostly envelop the inhibitory parvalbumin (PV) expressing interneurons. While the composition of PNNs may vary across different brain regions and change with age, their backbone is composed of chondroitin sulfate proteoglycans (CSPG) lecticans, such as aggrecan, brevican, and neurocan, whose amino terminal regions bind to hyaluronan while their carboxy terminal regions bind to tenascin [4,5]. The second CNS-specific form of the ECM is the basement membrane (BM), including the meningeal BM, which surrounds the pial surfaces, and the vascular BM, which surrounds blood vessels. The vascular BM is composed of laminin, collagen IV, fibronectin, and heparan sulfate proteoglycans and contributes to the integrity of the blood–brain barrier (BBB) [6,7]. Both neurons and glia contribute to the formation and maintenance of the ECM in the CNS [8].

During development, the ECM plays a role in the proliferation and differentiation of neuronal progenitors, dendritic and axonal growth and guidance, migration, cortical folding, connectivity, and synaptic plasticity [9,10,11]. For instance, the formation of PNNs seems to reduce synaptic plasticity after developmental critical periods of heightened neuroplasticity. Intriguingly, PNNs form and mature at different times in distinct brain regions during development. For instance, PNNs form at postnatal day 4 (PD 4) in the brainstem, at PD 14 in the cortex, and at PD 21 in the amygdala. The brain-region-specific timing of PNN development and maturation indicates that critical periods may also be brain-region-specific [10].

In the adult brain, the ECM regulates neuronal activity, in part by controlling the extracellular ion homeostasis, the expression of neurotransmitter (NT) receptors and ion channels, spine numbers, and spine maturity [12,13]. Neuronal activity can be altered by the binding of ECM components, such as reelin and fibronectin, to their receptors on neural cells via signaling pathways and kinases that increase activity of neurotransmitter receptors (i.e., NMDA) and voltage-gated calcium channels. A high negative charge of PNNs due to highly sulfated residues of glycosaminoglycans (GAG) can bind ions and signaling molecules, including growth factors, as well as provide neuroprotection against oxidative stress (Figure 1A). Because of its high hydration capacity, the ECM can also regulate the volume of the extracellular space and thereby control levels and diffusion of ions and neurotransmitters and, consequently, brain activity [8,14]. Vascular BM regulates which fluids and soluble molecules can enter and leave the brain [6].

There are many reports of ECM alterations, including degradation, overproduction, and altered composition, in brain conditions such as Alzheimer’s disease (AD), Huntington’s disease (HD), Parkinson’s disease (PD), schizophrenia, autism spectrum disorder (ASD), pain, and epilepsy [15,16,17,18,19]. The components of ECM that regulate its degradation are enzymes including matrix metalloproteinases (MMPs), disintegrin and metalloproteinase with thrombospondin type 1 motif (ADAMTs), hyaluronidases/chondroitinases, and plasmin, and their regulators including tissue inhibitors of metalloproteinases (TIMPs), tissue plasminogen activator (tPA), and plasminogen activator inhibitor. MMPs are a family of zinc-dependent endoproteases comprising 28 individual members involved in several physiological processes, including tissue morphogenesis and cell migration, and pathophysiological processes such as inflammation and cancer [20]. The activity of MMPs is regulated by their expression, processing, and the expression of their regulators. MMPs promote the turnover of collagens, elastin, gelatin, and other matrix glycoproteins and proteoglycans that compose the ECM. Additionally, MMPs directly degrade ECM proteins such as aggrecan, laminin, entactin, and fibronectin [21]. Importantly for PNNs, gelatinases such as MMP-2 and MMP-9 degrade aggrecan, laminin, and fibronectin [8]. MMPs can be released by a variety of cell types, including microglia [22]. The degradation of PNNs by MMPs in disease can contribute to PV interneuron dysfunction, and as a consequence alter the balance of excitation and inhibition in the affected brain regions. Loss of PNNs can also lead to reduced levels of growth factors, loss of neuroprotection against oxidative stress, altered concentration of ions in neuronal microenvironment, as well as perturbed expression of NT receptors and ion channels causing maladaptive neuroplasticity [23,24,25].

Recent advancements in ECM isolation and “omics” approaches have increased our understanding of proteins constituting the ECM and of the degree of similarity between human and mouse ECM [26]. For instance, proteomic analysis of mouse and human brain vascular ECM demonstrated a significant overlap, with 66% of human ECM markers present in mouse samples [26]. In contrast, RNA sequencing studies have revealed important differences in ECM expression between the developing human neocortex and the embryonic mouse neocortex, with significantly more ECM genes being expressed in the human fetal neocortex [27]. These studies indicate that using mouse models to investigate ECM may need to be interpreted with caution and that the use of human iPSCs ECM models or tissues is needed to translate findings.

In this brief review, we aim to discuss alterations in the expression of ECM genes in various brain disease states, the role of the transcriptional regulators HIF-1 and AhR in ECM gene expression, and the role microglia play in PNN regulation. We refer to other reviews for other aspects of ECM regulation and function [3,5,8,10,12,13,15,16,18,27,28,29,30].

## 2. Regulation of ECM Gene Expression in Development and Disease

Understanding ECM-related contributions to normal brain function and brain diseases requires the identification of ECM genes and pathways that regulate those functions and are altered in disease. Transcriptomic analysis has been a useful tool in many different fields, and genetic pathway analysis processes are becoming a common tool in the research field of neurodegeneration and disease (SNP, RNAseq, qPCR, etc.). This type of analysis sheds light on genes and related genetic pathways involved in normal development, function, and genes disrupted in disease pathology. Here, we summarize the findings from published works focusing on genetic pathway analysis of neurite outgrowth and brain diseases. This will provide a base understanding of the types of proteins or receptors that may have the greatest influence on the genetic regulation of the ECM.

### 2.1. ECM Genes Implicated in Neurite Outgrowth

The ECM plays an important role during the development of the nervous system [10,27]. Analysis involving dorsal root ganglia (DRG) has highlighted a few potential genes of interest when discussing the importance of the ECM in neurite outgrowth [31]. For instance, nociceptor DRG sensory neurons can be divided into two types depending on their ability to bind *Griffonia simplicifolia* isolectin B4 (IB4). Importantly, IB4+ DRG neurons have a decreased ability to regenerate neurites in vitro in comparison to IB4− DRGs that can regenerate neurites. Taking advantage of these differences, Fudge et al. wanted to identify factors permissive to neurite outgrowth by performing genetic pathway analysis [31]. They discovered a number of genes or genetic pathways that are involved in ECM nervous system development associated with the ability to extend neurites. In particular, a select group of nine genes was found to be differentially expressed between IB4+ and IB4- neurons; eight were found to be downregulated in the IB4+ neurons (*Icam1, Itgb1, Fn1, Spp1, Lamb1_predicted, Ctsh, Adamts1,* and *Plat*) and only one was found to be upregulated (*Plaur*). Of the downregulated genes, the general breakdown of gene functionality resides in ECM components fibronectin (*Fn1*) and laminins (*Lamb1*), both previously identified to play important roles in neurite outgrowth [32]. In addition, IB4+ neurons have reduced expression of genes encoding for proteins that regulate ECM degradation. These include *Adamts1*, encoding ADAMTS1, whose major substrates are lecticans, but which can also contribute to cell polarization and migration via the regulation of cellular Rho-GTPases activities [33,34,35]. *Cathepsin H* (*Ctsh*) encodes for serine protease, which was shown to increase the expression of MMP genes through the degradation of histone deacetylases (HDACs) [36]. Interestingly, *Plat* and *Plaur*, two genes involved in plasmin activation, are regulated in opposite directions. *Plasminogen activator* (*Plat*) encodes for a protein that catalyzes the plasminogen–plasmin conversion to modulate laminin degradation and activation of trophic factors such as nerve growth factors as well as activation of microglia. *Plaur* encodes for the receptor urokinase plasminogen (u-PAR), which, upon binding to the protease urokinase-type plasminogen activator (uPA or urokinase), initiates plasmin formation and promotes localized ECM degradation. As *Plaur* is expressed at higher levels and *Plat* at lower levels in the IB4+ DRG neurons, this suggests that fine-tuning of ECM degradation towards a “golden middle” with neither too little nor too much ECM remodeling could be essential for optimal neurite growth. In addition, a recent study indicated that the mechanosensation of the stiffness of CNS tissue influences neurite outgrowth [37]. As the ECM contributes to overall CNS tissue stiffness, this further indicates the importance of balancing ECM deposition and degradation in brain physiology. Furthermore, investigating whether the expression of genes that regulate ECM degradation and deposition is altered in brain diseases may provide insight into the origin of the imbalance of these processes and how it can contribute to disease pathogenesis.

### 2.2. ECM Genetic Pathway Disruptions in Disease

Genetic disruption of ECM pathways and genes is found in different brain conditions, including epilepsy, neuropathic pain, and neurodegenerative disease. Although perturbation of the ECM has not been shown to directly cause neurodegeneration, it can make neurons more susceptible to dysfunction and cell death [16].

#### 2.2.1. Epilepsy

Epilepsy is a common chronic neurological disease and there are no current treatments available to prevent the development of epilepsy. The process of epileptogenesis can be studied using animal models including the pilocarpine and kainic acid rat and mouse models. Pilocarpine is a muscarinic agonist commonly used to model epileptogenesis in rats, as after injection, rats will exhibit seizures that phenotypically resemble human temporal lobe epilepsy (TLE) [38]. Similarly, kainic acid (KA) is a potent neurotoxin that produces neuronal excitation, leading to similar TLE phenotypes in rodents [39]. Han et al. performed microarrays from pilocarpine and KA TLE animal models of epilepsy and identified a total of 567 differentially expressed genes (DEGs) shared between the two models [34]. Pathway analysis identified four major gene pathways involved in epileptogenesis. These pathways include “Focal adhesion”, “ECM-receptor interaction”, “Adherens junction”, and “Cell adhesion molecules (CAMs)”. Specifically, upregulation of seven genes (*Col6a3*, *Lamb2*, *Flna*, *Flnc*, *Itga1*, *Itga2b*, and *Itgb1*) was observed in the hippocampus of intra-amygdala KA-injected rats. These genes encode for ECM components collagen IV (*Col6a3*) and laminin B2 (*Lamb2*), integrin alpha and beta subunits (*Itga1*, *Itga2b*, and *Itgb1)* that function as ECM receptors, and filamins (*Flna*, *Flnc*) which are cellular proteins that connect cytoskeleton actin with integrins [40]. This phenomenon is mirrored in human patients with TLE whose hippocampal tissue also expressed higher levels of related genes such as *Itgb1* and *Flna* [34].

Although Han et al. found *Lamb2* to be upregulated in their epilepsy models, further research into the KA model of epilepsy suggested that laminin protein levels might play an important role in KA excitotoxicity [41]. They found that two hours after KA injections, neurons had an overall normal morphology, but laminin protein levels in the CA1 and CA3 regions of the hippocampus had nearly vanished. It was not until two days after injection that the neurons in the CA1, CA2, and CA3 had degenerated. Whether the upregulation of laminin-related genes mentioned above may be in an attempt to compensate for this protein degradation remains to be determined. The authors proposed that degradation of laminin precedes the neuronal loss in the regions that eventually experience TLE neuronal degeneration. The role of laminin in epileptogenesis seems to be an important factor in maintaining neuronal viability and potentially preventing neuronal degeneration.

A study by Dubey et al. provided insight into the importance of the PNN and MMPs in epileptogenesis [42]. Epilepsy and seizures are hypothesized to be the result of an imbalance in excitation and inhibition. As the PNN most prominently is involved in the organization of synaptic stability and GABAergic interneurons within the CNS, faulty PNN circuitry could play a role in the initiation and maintenance of seizure states [25]. It has been found that PNN integrity is lost, especially in the hippocampus, in animals and patients with chronic epilepsy. MMP-13 is expressed in the brain and is known to cleave aggrecan, a PNN-specific lectican [43]. MMP-9 and -13 have been found to increase following status epilepticus (SE) [44]. For instance, there was a 4-fold increase in MMP-13 mRNA levels two days post-SE and a 2-fold increase in the protein level one week post-SE. Dubey et al. found that MMP activity also increases following SE induced by methyl-scopolamine and pilocarpine injections. The increased level of MMP-13 protein was colocalized with PNN positive cells in the hippocampus and cortex, suggesting the role of MMPs in the degradation of the PNN in SE. Studies using MMP-9 knockout or MMP-9 overexpressing mice further support this as pentylenetetrazole (PTZ) kindling, a mechanism that causes seizures, is inhibited in MMP-9 knockout mice and is increased in MMP-9-overexpressing transgenic mice [45].

A recent study demonstrated a reduction in PNNs and significantly altered expression patterns of versican, neurocan, aggrecan, and WFA-specific glycosylation in the hippocampus of patients with drug-resistant mesial temporal lobe epilepsy (MTLE) [46].

#### 2.2.2. Neuropathic Pain

Neuropathic pain is another brain condition in which regulation of the ECM seems to play an important role. While many people suffer from chronic neuropathic pain, its underlying mechanisms remain poorly understood. As nerve injury and inflammation of the nervous system can cause altered gene expression in neuronal tissue, it is thought that these long-lasting changes in gene expression can contribute to developing neuropathic pain. Intriguingly, pathway analysis of perturbed genes in two mouse models of pain (nerve injury and inflammation-induced pain) identified ECM organization as the most commonly regulated pathway across the tissues tested [47].

The altered pathways contain genes essential for the biological processes that regulate assembly, maintenance, and disassembly of the ECM, including genes encoding for different collagens (*Col5a3*, *Col1a1*), matrix metallopeptidase 13 (*Mmp13*), and other ECM-related proteins (*Comp*, *Ctss*, *Sparc*, *Vwf*, and *Thbs1*). Secreted and rich in cysteine and acidic amino acids (*Sparc*) and thrombospondin 1 (*Thbs1*) are known for their role in synapse formation [48]. While a majority of these genes are upregulated in the mouse models of neuropathic pain, interestingly, the cartilage oligomeric matrix protein (*Comp)* was downregulated in both mouse models (specifically found in dorsal root ganglia and spinal cord tissue samples). *Comp* is a large glycoprotein that interacts with multiple ECM proteins in cartilage and other tissues [49]. The altered regulation of *Comp* is known to contribute to pathology in many disorders such as fibrosis, cardiomyopathy, and arthritis, yet the mechanisms leading to said dysregulation are not well studied. The prevalence of *Comp* dysfunction in these various conditions may suggest that this ECM protein is an important factor in disease states. Furthermore, given that dysregulation of genes within the ECM organization pathway was conserved between two models of pain, it is possible that ECM dysregulation could be a common theme contributing to the development of chronic pain.

#### 2.2.3. Cerebellar Ataxia

ECM dysregulation in neurodegenerative diseases, including Alzheimer’s disease (AD), adult-onset leukoencephalopathy with axonal spheroids and pigmented glia (ALSP), and Huntington’s disease (HD), has been previously described, indicating that ECM perturbation may contribute to disease pathogenesis in different regions of the cerebrum affected in these diseases. Less is known about ECM dysfunction in the cerebellum, the brain region that contains the majority of the neurons in the human brain. Spinocerebellar ataxia type 1 (SCA1) is a dominantly inherited neurodegenerative disorder caused by the expansion of CAG repeats in the *ATAXIN-1* (*ATXN1*) gene that is characterized by early and severe pathology in the cerebellum [50,51]. Repeat expansions of 39 or more consecutive CAG repeats results in the onset of disease in SCA1. Symptoms include loss of balance and coordination, impairments in cognition and mood, and premature death [52]. Researchers studying SCA1 commonly use mouse models to examine underlying molecular mechanisms of disease development and progression [53,54]. One such model is a knock-in mouse model in which 154 CAG repeats were inserted into the mouse *Atxn1* gene referred to as *Atxn1*^154Q/2Q^ [55]. Given that in the developing lung, the ATXN1 family regulates ECM remodeling, it is possible that ATXN1 might have a similar function in ECM regulation in the cerebellum and that mutations in ATXN1 could result in dysregulation of ECM genes [56]. Indeed, RNAseq on cerebellar cortex tissue identified DEGs involved in ECM regulation [57]. These included genes encoding ECM components collagen (*Col4a5*, *Col6a5, Col11a1, Col9a2*), laminin (*Lamc3*), integrins (*Itga3, Itgb3, 5 and 6*), and ECM remodeling (*Adamts10, Adamts4, MMP16, MMP17*). We also found altered expression of ECM genes regulating synapses including *Sparcl1*, *Thbs2*, and *cerebellin 1* (*Cbln1*). KEGG pathway analysis identified perturbation in ECM pathways that included “collagen-containing extracellular matrix”, “hemoglobin complex interactions”, “synaptic membrane”, and “transporter complexes”.

Cerebellin-1 (*Cbln1*) is an ECM glycoprotein involved in cell adhesion that may play a role in cerebellar ataxia. *Cbln1* is released from the parallel fibers of cerebellar granule cells and has been shown to play a role in the synaptic organization of the cerebellum [58]. In particular, *Cbln1* can bind neurexins to the amino terminal domains of GluD2 (glutamate receptor family member delta-2) on Purkinje cells, promoting synapse creation between parallel fibers and Purkinje cells [59]. *Cbln1*-null mice display motor deficits similar to that of cerebellar ataxia and Suzuki et al. demonstrated that these motor deficits could be ameliorated in adult mice using the synthetic synaptic organizer protein CPTX [60]. CPTX utilizes the modular architecture of *Cbln1* and a neuronal pentraxin (NP1) in an attempt to form excitatory synapses by physically bridging pre- and postsynaptic sites. After injecting CPTX into cerebellar lobules VI and VII of adult *Cbln1*-null mice, they found that CPTX treatment partially restored the synapses between parallel fibers and Purkinje cells and improved motor deficits of injected *Cbln1*-null mice. This effect decayed after the initial injection, which suggests that continued presence of CPTX is necessary for synaptic maintenance. 

These studies indicate that disruptions in ECM maintenance may underlie the dysfunction in the SCAs. Regulation of the ECM contributes to neuronal capabilities since it provides the scaffold for various structures critical for forming and maintaining effective connections between neurons. These include neuronal outgrowth and maintenance as well as ECM components involved in synaptic plasticity, formation, and regulation [29]. Prolonged disruption of these pathways could lead to disruptions in proper synapse formation and transmission. Synaptic pathology within the CNS has been noted in SCA1 mouse models, one of the most prominent being the altered climbing fiber innervation seen in cerebellar Purkinje cells [61]. This synaptic alteration is accompanied by notable dendritic atrophy as the disease progresses. Decreased expression of *Cbln1* and dysregulation of other ECM genes may contribute to the loss of synapses and dendritic atrophy in SCA1 cerebella.

#### 2.2.4. Age-Related ECM Changes and Neurodegeneration

Aging is an important risk factor in most neurodegenerative diseases. One way by which aging can contribute to neurodegenerative diseases is via age-induced ECM degradation. In addition to causing an imbalance between ECM degradation and deposition, degradation of the ECM produces signaling peptides, such as elastin-derived peptides (EDPs) [62]. Elastin is an important component of the ECM, particularly in vascular BM, where it mediates the elasticity of arteries. Elastin has extraordinary stability and an extremely low turnover rate; radiocarbon prevalence data indicate that elastin in the lungs is the same age as the human subjects [63]. This indicates that elastin degradation is potentially irreversible or irreparable in aging and neurodegenerative or pathophysiological conditions [64]. While being generally resistant to proteolysis, elastin can be degraded by proteinases such as MMPs. Moreover, when elastin degrades, newer peptides form called elastin-derived peptides (EDPs), and the levels of EDPs increase with aging [65]. While still largely understudied, the increased presence of EDPs with age supports the idea that they play an important role in the progression and development of age-related disease disorders [66]. Importantly, when EDP binds to an elastin-binding protein (EBP) receptor, it activates ERK and AKT kinases. This can result in increased expression of *Mmps*, which can then further degrade the ECM and promote inflammation [67]. In addition, elastin-like polypeptides (ELPs) can induce the expression of genes such as *Beta secretase1 (BACE1*) that can directly contribute to the deposition of amyloid beta plaques, a hallmark of AD [68]. The processing of amyloid precursor protein (APP) into amyloid beta (Aβ) leads to the formation of Aβ plaques in the brain. Ma et al. showed that ELPs can induce overproduction of Aβ in a Chinese hamster ovary cell line that stably expresses mutant human amyloid precursor protein (7PA2 cells). They also found that ELP injections in wild-type C57BL/6 mice caused a significant upregulation of Aβ production. Treatment with ELPs in mice showed more than just upregulation of Aβ levels; these mice exhibited both pathological and neurobehavioral AD phenotypes, confirming a relationship between ECM degradation and pathogenesis of AD. Together these studies suggest that elastin in brain ECM may be an important factor in initiating the neurodegenerative deficits and pathological changes due to AD progression. 

In summary, these studies indicate that regulation of the ECM gene expression is necessary for normal brain function and that dysregulated expression of ECM genes is a common feature of CNS disease. Here we discussed gene expression changes in ECM components that regulate the balance between deposition and degradation and how disbalance of these processes contributes to pathogenesis of epilepsy, neuropathic pain induced by nerve injury and inflammation, cerebellar ataxias, and age-related disorders such as AD. Further studies are needed to identify the mechanisms of these transcriptional perturbations, including the identification of key transcriptional regulators.

## 3. Transcriptional Regulation of ECM: HIF-1, AhR, and a Potential Role in ECM Regulation

A common link between aging, the most common risk factor in neurodegeneration, and genetically diverse neurodegenerative disorders is exposure to periods of low oxygen availability called hypoxia. For example, obstructive sleep apnea (OSA) is a type of sleep-disordered breathing characterized by recurring periods of upper airway obstruction that disrupt air flow and cause intermittent hypoxia. The prevalence of OSA increases with age and OSA is implicated in numerous neurological disorders including AD, PD, multiple system atrophy (MSA), amyotrophic lateral sclerosis (ALS), epilepsy, and several hereditary ataxias [69,70,71,72,73,74]. Sleep-disordered breathing is associated with cognitive dysfunction and increased mortality [75,76,77]. In addition, age-induced vascular changes can contribute to hypoxia. For instance, the degradation of elastin, which constitutes 50% of dry arterial wall weight, contributes to arterial stiffness and hypertension, which can contribute to hypoxia [67].

Hypoxia-inducible factor 1 (HIF-1) is a dimeric transcription factor composed of a β subunit and an α subunit that is inhibited by HIF prolyl hydroxylases (PHD) in normoxia. Hypoxia causes the destabilization of HIF-1α inhibitors and HIF-1 activation to elicit changes in gene expression. HIF-1 targets a variety of genes that help cells adapt to low oxygen levels, including erythropoietin (EPO), vascular endothelial growth factor (VEGF), and genes involved in metabolism. These pathways are hypothesized to elicit neuroprotective effects by promoting angiogenesis, blood flow, oxygen delivery, metabolism, and redox balance. HIF-1 also targets genes involved in ECM remodeling, including collagen synthesis and some metalloproteinases, suggesting that hypoxia could contribute to ECM remodeling [78]. Studies that examine the role of hypoxia and HIF-1 in promoting metastasis in cancer are examples of how HIF-1 activation can regulate the expression of ECM components including MMPs to alter ECM physiology [79]. Even a single hypoxic event can induce long-lasting changes in the ECM, as demonstrated in perinatal rats which exhibited altered PNN composition in the cingulate cortex that persisted into adulthood from exposure to a single hypoxic event [80]. While some studies show a neuroprotective role for HIF-1 in neurodegeneration and brain injury, others suggest that HIF-1α might contribute to pathology [81]. It seems that HIF-1 has dual transcriptional activity and can promote cell survival or cell death depending on variables such as the magnitude, duration, and frequency of hypoxia, as well as the cell type with HIF-1 activity. Here we discuss cell-type-specific functions of HIF-1 and a potential role for glial HIF-1 in mediating the transcriptomic and physiological disruptions of the ECM frequently observed in disease. 

HIF-1 signaling occurs in neurons, glia, and vascular cells of the brain and may have cell-type-specific effects on transcriptional activity and cell viability. Glial cells, including astrocytes and microglia, have a critical role in supporting neuronal function and homeostasis. Astrocytes function to maintain the extracellular milieu, regulate the blood–brain barrier (BBB), and regulate neuroinflammation. Vangeison et al. showed that HIF-1α loss of function (LOF) differentially affects neuronal survival depending on if astrocytes or neurons are targeted. HIF-1α LOF in neurons reduced neuronal survival after moderate hypoxia, while HIF-1α loss in astrocytes had a neuroprotective effect in vitro. The deleterious effects of HIF-1α expression in astrocytes on neuronal viability might be due to the upregulation of inflammatory agents, as observed by the increase in *inducible nitric acid synthase* (*iNOS*) mRNA and the dose-dependent rescue in cell viability from iNOS inhibitors [82]. Another example of HIF-1 cell-type-specific activity involves pericytes, which have important roles in regulating the BBB. Baumann et al. used a similar HIF-1α LOF paradigm to examine the role of pericytic and astrocytic HIF-1α signaling in vascular permeability in an in vivo model. They found that HIF-1α loss of function in pericytes, but not astrocytes, ameliorates enlarged vessels and increases BBB permeability induced by hypoxia. Hypoxia-induced BBB permeability is thought to stem from HIF-1α-induced expression of VEGF, TGF-β, and MMPs, but intriguingly, pericytic HIF-1α LOF did not reduce the expression of VEGF, TGF-β, MMP-2, or MMP-9, suggesting that pericytic HIF-1 signaling may impact BBB integrity through alternate mechanisms [83]. Additional support for neuronal HIF-1 having a beneficial effect on neuronal viability is demonstrated in experiments where HIF-1α LOF in sensory neurons impaired axonal regeneration in vitro and in a mouse model of nerve lesion, while the induction of HIF-1α by acute intermittent hypoxia enhanced regeneration in a mechanism involving known HIF-1 targets including *VEGF* [84].

Microarray experiments from astrocyte cell cultures exposed to hypoxia demonstrate an upregulation of HIF-1α signaling pathways, interleukin gene expression, chemokine expression, and upregulation in the expression of many matrix metalloproteinase genes (including MMP-9 and MMP-13), suggesting that hypoxic astrocytes could contribute to ECM regulation [85]. Hypoxic stimulation of primary rat astrocyte cultures increased mRNA and protein levels of MMP-13, and rat brain endothelial cells exposed to the hypoxia-conditioned astrocyte media exhibited increased permeability [86]. Increased levels of the proinflammatory molecules IL-1 and TNF-α in astrocytes regulate gelatinase MMP levels [87]. IL-1β has a hypoxia response element, suggesting that HIF-1 could directly mediate IL-1β expression and hypoxic astrocytes release IL-1β in vitro in a HIF-1 dependent manner [88].

Microglia are the resident immune cells of the brain that survey their environment and perform physiological housekeeping roles [89]. Microglia become activated during injury and inflammation, during which they release a variety of proinflammatory molecules and proteases that regulate the ECM, including MMPs [90]. The effect of hypoxia on microglia appears to have a similar effect as in astrocytes by upregulating inflammatory pathways. Zhang et al. showed that acute hypoxia led to a shift from anti-inflammatory to pro-inflammatory microglial phenotypes and was associated with an upregulation of the NF-κB pathway in a mouse model of Alzheimer’s disease [91]. Furthermore, they observed that hypoxia leads to an increase in the mRNA of chemokines (*CCL2*, *CCR2*, and *CCL3)*. Given that NF-κB is a known regulator of HIF-1α in hypoxia and *CCL2* is a known target gene of HIF-1α, HIF-1α might be involved in this proinflammatory process [92]. Rats exposed to intermittent hypoxia (IMH) exhibited increased expression of *iNOS*, *TNFα*, and *IL-6* mRNA in microglial cells and chronic IMH increased *TLR-4* mRNA levels, which were associated with increased *HIF-1α* mRNA [93]. 

Like HIF-1, the Aryl hydrocarbon receptor (AhR) is a transcription factor implicated in aging, neurodegenerative disease, and ECM regulation. AhR and HIF-1 are both members of the basic helix–loop–helix/PER–ARNT–SIM (bHLH-PAS) protein family. These pathways have a complex relationship such that their simultaneous activation can have enhancing or antagonistic effects on downstream gene targets depending on the specific cell type and type of exposure [94,95]. AhR activity could also indirectly increase HIF-1 activity by decreasing the expression of PHD to disinhibit the hypoxia-dependent α subunit [96]. AhR is activated by xenobiotic compounds such as dioxins that are released into the environment by industrial activities. AhR ligand binding causes a conformational change that exposes a nuclear localization signal which then allows for AhR translocation into the nucleus. Upon nuclear entry, AhR dimerizes with HIF-1β, also known as Aryl hydrocarbon receptor nuclear translocator (ARNT), and interacts with xenobiotic response elements to modulate gene expression. For example, 2,3,7,8-tetrachlorodibenzo-p-dioxin (TCDD) is an environmental chemical that can bind AhR, allowing it to enter the nucleus and stimulate the expression of metabolizing genes such as cytochrome p450 1 family members [97]. In the neurodegenerative diseases PD and HD, AhR may contribute to pathogenesis by altering the expression of ECM genes including MMPs, TIMPS, and genes involved in cell adhesion, cell–cell or cell–matrix interactions, as well as by promoting neuroinflammation [96,98]. In a mouse model of liver fibrosis, TCDD AhR activation increased the expression of MMP-3, MMP-8, MMP-9, MMP-13, TIMP-1, plasminogen activator genes, and procollagen genes [99]. In human orbital fibroblasts, AhR activation by the AhR ligand 6-formylindolo [3,2b]carbazole (FICZ) led to the expression of MMP-1 in an AhR- and ARNT-dependent manner and decreased collagen levels in a fibrosis model [100]. This evidence supports a role for AhR in ECM regulation, but further research is required to understand how AhR signaling alters ECM in the brain and how ECM gene regulation is impacted by simultaneous AhR and HIF-1α activity. 

Altogether, this suggests that hypoxia can promote ECM dysregulation directly through its transcriptional activity on ECM components and indirectly in a mechanism that involves inflammation which promotes glial release of ECM-modifying proteases such as MMPs. The increased expression of MMPs could contribute to pathology and explain selective cell death in neurodegenerative disease. In a mouse model of ALS, reducing MMP-9 levels ameliorated neuronal loss and the expression of inflammatory proteins while extending survival [101]. Moreover, a transcriptomic study that assessed DEGs between spared and vulnerable motor neuron populations in a mouse model of ALS found that MMP-9 was strongly expressed in vulnerable but not spared motor neurons and that reducing MMP-9 levels improved pathology [102]. Hypoxia, neuroinflammation, and ECM dysregulation are all common themes in neurodegenerative disease, and it is possible that HIF-1α transcriptional activity could be a common mediator of the changes in the ECM in these conditions. To understand how hypoxia and HIF-1 alter the ECM, it is important to understand the crosstalk between HIF-1 and AhR. Moreover, another level of complexity arises from the fact that HIF-1β may also be regulated by hypoxia [103]. Additional investigation concerning whether HIF-1β expression is altered by hypoxia in neural cell types and how mechanistically this occurs is necessary to understand the extent to which HIF-1 activity may alter the ECM. Further research is required to understand the cell types involved in hypoxia-induced ECM regulation, how ECM dysregulation may contribute to disease, and how HIF-1 could be a therapeutic target to ameliorate changes in the ECM.

## 4. Regulation of Perineuronal Nets by Microglia

Thus far, we have discussed research that has used “-omic” approaches in bulk tissues. As noted in previous sections, many cell types in the brain can express ECM genes. Due to their active role in phagocytosis and reactive changes with aging and in disease, here we focus on a few studies investigating the role of microglia in regulating the balance of ECM degradation and deposition, and in particular, brain-specific ECM component perineuronal nets. These studies provide insight into how microglia regulate PNNs and how microglial dysregulation of PNNs may contribute to neurodegenerative disease. We recommend other reviews for a more in-depth overview of microglia and ECM [8,104,105].

PNNs are brain-specific ECM structures that envelop various neuron types within cortical regions during the closure of critical periods. PNNs consist of hyaluronan, chondroitin sulfate proteoglycans (CSPGs), tenascin, hyaluronan and proteoglycan binding link protein, and glycosaminoglycans (GAGs) polysaccharides [5,22]. Primarily, PNNs form around parvalbumin-positive (PV) inhibitory interneurons, serving to stabilize synaptic connectivity [106]. The importance of PNNs for brain function was demonstrated by studies where the removal of CSPGs, a PNN component, resulted in increased long-term memory, disrupted contextual fear, and sensitized pain [23,106,107].

Microglia play a role in the homeostasis of PNNs, as was demonstrated in a recent study by Nguyen et al. [108]. The authors showed that an enriched environment promotes IL-33 production by hippocampal neurons; promotes microglia-dependent ECM remodeling, hippocampal dendritic spine formation, and synapse plasticity; and is required for memory consolidation. IL-33 induces changes in microglial gene expression, including *Gas7*, a gene that plays a role in phagocytosis, and *Marco*, a gene that plays a role in debris clearance. This indicates that IL-33 can induce ECM phagocytosis by microglia. Moreover, super-resolution imaging and 3D reconstruction revealed that IL-33 KO mice exhibited reduced levels of the PNN component aggrecan within microglial lysosomes, indicating that IL-33 promotes microglial phagocytosis of PNN during learning. In addition, IL-33 induced the expression of genes involved in ECM degradation, including *Mmp14*, *Mmp25*, and *Adamts4*, indicating that microglia also promote ECM degradation by secreting these enzymes. Importantly, this IL-33-induced and microglia-mediated remodeling of ECM was required for long-term contextual fear memory. Intriguingly, microglia, ECM clearance, and IL-33 are all implicated in aging and mouse models of AD, suggesting that dysregulation of ECM clearance by microglia may contribute to age and AD cognitive decline [108].

Another study by Strackeljan et al. investigated the effects of reversible pharmacological microglial depletion on remodeling of PNNs in the CA1 region of the hippocampus, a region predominantly involved in learning [109]. Microglia depletion with PLX, an inhibitor of colony-stimulating factor 1 receptor (CSF1R) signaling, decreased the size of PNN holes and elevated the expression of the surrounding ECM. Furthermore, microglia depletion increased the expression of perisynaptic ECM component brevican and the number of excitatory presynaptic puncta labeled with vesicular glutamate transporter 1 (VGLUT1). This provides further evidence that in healthy brains, microglia regulate PNNs and could potentially alter the physiology of PNN positive neurons in regions associated with memory.

Additional studies indicate that microglia-mediated alterations of PNNs contribute to the pathogenesis of neurodegenerative diseases such as AD, HD, and adult-onset leukoencephalopathy with axonal spheroids and pigmented glia (ALSP) [110,111,112]. Reactive microglia are a shared feature amongst neurodegenerative diseases [113,114]. Here, we outline a few studies describing abnormal microglial phagocytosis of PNNs as well as the release of proteases that degrade PNNs in these diseases.

ALSP is an autosomal dominant neurodegenerative disease caused by mutations in the kinase domain of *colony-stimulating factor 1 receptor* (*CSF1R*) primarily expressed by microglia. ALSP-causing mutations seem to eliminate the kinase activity of CSF1R, and *Csf1r* haploinsufficiency alone could cause ALSP. This led to the development of global CSF1R^+/−^ mice as a model of ALSP [112]. As CSF1R is also expressed in neurons and is important for neuronal development and survival, Arreola et al. used a conditional mouse genetic approach to delete one allele of *Csf1r* only in microglia (*Cx3cr1-CreERT2/+; Csf1r+/fl* mice) to determine the effects of microglial CSF1R haploinsufficiency. While CSF1R is important for microglial survival, a 50% reduction in its expression in microglia did not affect the density of microglia, but did decrease the expression of microglial homeostatic genes. This dyshomeostasis of microglia was associated with disruption of PNNs similar to those seen in the global CSF1R^+/−^ mouse model of ALSP. This indicates that microglial dyshomeostasis caused by the loss of one *Csf1r* allele in microglia is sufficient to perturb the remodeling of PNNs. To further test the role of microglial dyshomeostasis in ALSP, authors depleted microglia in the global CSF1R^+/−^ mouse model by pharmacologically inhibiting CSF1R signaling with low doses of inhibitor PLX5622. PLX treatment elicited a robust recovery of PNN markers including *Wisteria floribunda* Agglutinin (WFA) and aggrecan in the cortex, and rescued cognitive deficits in CSF1R^+/−^ mice [50]. These findings further support the claim that microglial dysfunction may cause the loss of PNNs and contribute to behavioral deficits in ALSP.

Alzheimer’s is another neurodegenerative disease that shows a reduction in PNN. Using a mouse model of AD (5xFAD) and tissue samples from patients with AD, Crasper et al. demonstrated progressive activation of microglia and reduction in PNN. Moreover, they show a close association between altered PNN and microglia, as well as the presence of PNN components within microglia, indicating direct phagocytosis of PNNs by reactive microglia. To directly test the role of microglia in AD PNN degradation, the authors used PLX5622 administration to deplete microglia. In two different mouse models of AD (5xFAD and 3xTg-AD), microglial depletion prevented PNN loss in AD mouse models [110]. While this study further suggests that microglial-mediated degradation of PNN is a feature of neurodegenerative diseases, how the loss of PNN contributes to the pathogenesis of AD remains to be explored. The role of microglia-mediated reduction in PNN and its contribution to disease phenotypes was demonstrated in the transgenic R6/2 mouse model of Huntington’s disease. In this mouse model, the depletion of microglia with PLX rescued the degradation of PNNs in the motor, somatosensory, and piriform cortex. This coincided with the rescue of striatal volume loss as well as the rescue of behavioral deficits (i.e., grip strength and novel object memory), indicating that microglia degradation of PNNs may contribute to striatal volume loss and HD phenotypes [111].

One mechanism by which reactive microglia promote the degradation of PNNs is via releasing proteases [90]. A recent in vitro study elucidates this through the use of microglial and neuronal cell cultures, and microglial reactive gliosis via poly I:C exposure. Microglia activated with poly I:C expressed and released ECM proteases MMP-2 and MMP-9. The addition of the poly I:C activated microglia-conditioned media to established hippocampal neuron cell cultures resulted in decreased levels of aggrecan and alterations in both excitatory and inhibitory presynaptic components in PNN positive neurons but not in PNN negative neurons [115]. This study provides further support to microglial capabilities to release PNN-reducing components in disease conditions [91].

Another mechanism of PNN reduction by microglia is phagocytosis. PNN reduction mediated by microglia phagocytosis is implicated in the spinal cord to produce a pain phenotype. PNNs within the spinal cord are found around large somas of spinoparabrachial projecting neurons in Lamina I of the dorsal horn. Three days after spared nerve injury (SNI), the intensity of WFA staining was significantly reduced around these neurons [116]. WFA signal was detected within microglial lysosomes, indicating that microglia are phagocytosing PNN. Importantly, ablation of microglia rescued both the PNN degradation (measured as a loss of WFA) and pain phenotypes (such as increased grimacing), suggesting that microglia-mediated degradation of PNNs on spinoparabrachial projection neurons promotes pain behavior [107].

While we have chosen here to focus on the impact of microglia on PNNs, it is important to note that they are not the only neural cell type that produces and regulates the ECM. For instance, PV interneurons themselves can impact PNNs. A recent study using chemogenetic approaches revealed that inhibition of PV interneuron activity is enough to cause a reduction in their PNNs [117]. As microglia, astrocytes, interneurons, and other neural cell types can affect PNNs, differentiating the extent of neuronal and glial contribution to the regulation of PNNs in healthy and diseased CNS is vital. This is especially important for in vivo neurodegenerative models where both interneurons and microglia are impacted.

In conclusion, several studies have revealed the relationship between microglia and PNNs, demonstrating that microglia release PNN-degrading proteins, phagocytose PNN components, and physiologically remodel PNNs. In neurodegeneration, activated microglia are involved in PNN degradation, contributing to important aspects of pathology, as the removal of microglia rescues or reverses some disease phenotypes. This concept of microglia as essential regulators of PNNs is further bolstered by observed disruptions in PNNs within the healthy brain with the modulation of microglia.

## 5. Summary

Despite being a significant part of the CNS, our understanding of the ECM’s regulation and role in brain physiology and pathology remains incomplete. However, the number of studies implicating the dynamic nature and regulation of ECM is steadily increasing. We focused here on studies investigating the transcriptional regulation of ECM genes as well as novel roles of microglia in the degradation of ECM components.

Transcriptional regulation and roles of ECM during development are well characterized and intuitive. Recent studies indicate that transcriptional perturbations of the ECM are a common feature of neurodegenerative diseases as well. While increased expression of enzymes degrading ECM components may exacerbate pathogenesis and contribute to selective neuronal vulnerability in neurodegeneration, how exactly the altered expression of structural components of ECM contributes to brain disease remains to be explored. Creating a database of ECM-related genes in neurodegenerative disorders can help provide a starting point for those researching the role of ECM degeneration in said disorders.

The identification of HIF-1α as a transcriptional factor regulating ECM gene expression may provide a shared molecular mechanism of ECM perturbation across neurodegenerative diseases and their major risk factor, aging. The complexity of HIF-1α cell-type-specific roles and crosstalk with the AhR pathway requires further investigation to understand the potential of HIF-1α as a therapeutic target for ECM normalization in disease. Cell-type-specific regulation of the ECM is further supported by several studies which have led to an increased appreciation for the role that transcriptional changes within microglia play in regulating the ECM in disease conditions.

Finally, we discuss the role of microglia in PNN regulation. Recent studies have shown that microglia impact PNNs in learning and in neurodegenerative disease. Activated microglia in disease impact PNNs through phagocytosis and release of PNN-degrading proteins. Although we have covered some impacts of this interaction, further neurodegenerative investigations should prioritize exploring behavior and the impact on neuronal activity and degeneration.

The majority of ECM studies described above have been performed in rodents. The gene and protein differences between the human and rodent ECM indicate a need for future studies to validate these findings in human ECM.

## Figures and Tables

**Figure 1 ijms-24-07049-f001:**
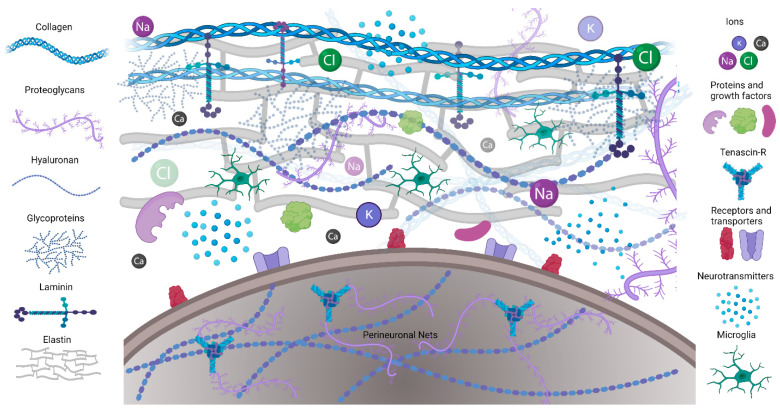
Schematic illustrating the ECM composition. Extracellular matrix proteins including collagen, elastin, and laminin provide structural support and form microenvironments for neuronal and glial interactions in CNS. Perineuronal nets are formed on the cell bodies and processes of mostly inhibitory neurons.

## Data Availability

Not applicable.

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
