# Peer review of "Extracellular Matrix Regulation in Physiology and in Brain Disease"

_ijms, 2023, doi:10.3390/ijms24087049_

Round 1
Reviewer 1 Report
1. In the introductory part, a figure is shown which does not represent different forms of ECM in the CNS – everything is mixed together and hyaluronan-based form of ECM (PNN) is not clearly visible. The changes in ECM in different conditions are shown in different colors but there is no explanation what that means and no references to the schemes in the text. Physiological functions of ECM are badly described, in particular those regulating synaptic plasticity and excitability.
2. The authors describe changes in individual ECM-related genes under different conditions and then “jump” to a conclusion on their functional relevance without involving in-depth discussion of data on the known functions of these molecules (e.g. from knockout and knockdown studies), their redistribution at the protein level, and not illustrating possible signaling pathways in which these molecules are involved.
3. The discussion of changes in ECM in epilepsy does not refer to published data showing changes in perineuronal net components (e.g. from Brenda Porter’s group).
4. Discussing the role of ECM in ataxia, one should mention the role of cerebellin (e.g. Suzuki et al., Science, 2020).
5. In several places the authors refer to some changes in MMPs and claim that this would result in ECM changes. This is too superficial. They need to introduce major MMPs and other extracellular proteases digesting BM or PNN components in the introduction and then refer to changes in these specific proteases and their possible effects on specific ECM forms and functions.
6. There was a recent review on “A glial perspective on the extracellular matrix and perineuronal net remodeling in the central nervous system”. Thus, the discussion of microglia – ECM interaction appeared not an original issue. It also lacks in-depth discussion of downstream events after ECM remodelling by microglia.
Minor points
Lines 87, 329, a period is missing
Line 146, microarray analysis
Line 181, collagens rather than collagen receptors?
Line 332, a fraction
Line 380, microglia
Author Response
We thank the reviewers for their helpful comments and suggestions that have improved our manuscript.
Our point to point responses (in blue) to reviewer’s comments (in italics) are bellow.
Reviewer 1
- In the introductory part, a figure is shown which does not represent different forms of ECM in the CNS – everything is mixed together and hyaluronan-based form of ECM (PNN) is not clearly visible. The changes in ECM in different conditions are shown in different colors but there is no explanation what that means and no references to the schemes in the text. Physiological functions of ECM are badly described, in particular those regulating synaptic plasticity and excitability.
Based on the reviewer’s comments we have revised Figure 1 to include a representation of PNNs, and have removed the rest of the figure for simplicity.
We have also added a brief description about ECM regulation of neuronal activity:
In the adult brain, the ECM regulates neuronal activity, in part by controlling the extracellular ion homeostasis, the expression of neurotransmitter (NT) receptors and ion channels, spine numbers, and spine maturity [12][13]. Neuronal activity can be altered by binding of ECM components, such as reelin and fibronectin, to their receptors on neural cells via signaling pathways and kinases that increase activity of neurotransmitter receptors (i.e. NMDA) and voltage gated calcium channels. A high negative charge of PNNs due to highly sulfated residues of glycosaminoglycans (GAG) can bind ions and signaling molecules, including growth factors, as well as provide neuroprotection against oxidative stress (Figure 1A). Because of its high hydration capacity, the ECM can also regulate the volume of the extracellular space and thereby control diffusion of ions and neurotransmitters and consequently brain activity [8][14].
- The authors describe changes in individual ECM-related genes under different conditions and then “jump” to a conclusion on their functional relevance without involving in-depth discussion of data on the known functions of these molecules (e.g. from knockout and knockdown studies), their redistribution at the protein level, and not illustrating possible signaling pathways in which these molecules are involved.
To address the reviewer’s suggestion we have thoroughly revised the manuscript including the following paragraph:
Of the downregulated genes, the general breakdown of gene functionality resides in ECM components fibronectin (Fn1) and laminins (Lamb1), both previously identified to play important roles in neurite outgrowth [32]. In addition, IB4+ neurons have reduced expression of genes encoding for proteins that regulate ECM degradation. These include a disintegrin and metalloproteinase with thrombospondin type 1 motif (Adamts1), whose major substrates are lecticans, but are also known to contribute to cell polarization and migration via regulation of cellular Rho-GTPases activities [33][34][35]. Cathepsin H (Ctsh) encodes for serine protease that was shown to increase the expression of MMP genes through the degradation of histone deacetylases (HDACs)[36]. Interestingly Plat and Plaur, two genes involved in plasmin activation, are regulated in opposite directions. Plasminogen activator (Plat) encodes for a protein that catalyzes the plasminogen-plasmin conversion to modulate laminin degradation and activation of trophic factors such as nerve growth factors as well as activation of microglia. Plaur encodes for the receptor urokinase plasminogen (u-PAR), which upon binding to the protease urokinase-type plasminogen activator (uPA or urokinase) initiates plasmin formation and promotes localized ECM degradation. As Plaur is expressed at higher levels and Plat at lower levels in the IB4+ DRG neurons, this suggests that fine-tuning of ECM degradation towards a “golden middle” with neither too little nor too much ECM remodeling could be essential for optimal neurite growth. In addition, a recent study indicated that mechanosensation of the stiffness of CNS tissue influences neurite outgrowth [37]. As the ECM contributes to overall CNS tissue stiffness, this further indicates the importance of balancing ECM deposition and degradation in brain physiology.
- The discussion of changes in ECM in epilepsy does not refer to published data showing changes in perineuronal net components (e.g. from Brenda Porter’s group).
We thank the reviewer for this excellent suggestion, following paragraph referring to data showing changes in PNN components is included in the revised manuscript.
A study by Dubey et al. provided insight into the importance of the PNN and MMPs in epileptogenesis [42]. Epilepsy and seizures are hypothesized to be the result of an imbalance in excitation and inhibition. As the PNN most prominently is involved in the organization of synaptic stability and GABAergic interneurons within the CNS, faulty PNN circuitry could play a role in the initiation and maintenance of seizure states [25]. It has been found that PNN integrity is lost, especially in the hippocampus, in animals and patients with chronic epilepsy. MMP-13 is expressed in the brain and is known to cleave aggrecan, a PNN specific lectican [43]. MMP-9 and -13 have been found to increase following status epilepticus (SE) [44]. For instance, there was a 4-fold increase in MMP-13 mRNA levels two days post-SE and a 2-fold increase in the protein level one week post-SE. Dubey et al. found that MMP activity also increases following SE induced by methyl-scopolamine and pilocarpine injections. The increased level of MMP-13 protein was colocalized with PNN positive cells in the hippocampus and cortex suggesting the role of MMPs in the degradation of the PNN in SE. Studies using MMP-9 knockout or MMP-9 overexpressing mice further support this as pentylenetetrazole (PTZ) kindling, a mechanism that causes seizures, is inhibited in MMP-9 knockout mice and is increased in MMP-9 overexpressing transgenic mice [45].
- Discussing the role of ECM in ataxia, one should mention the role of cerebellin (e.g. Suzuki et al., Science, 2020).
We thank the reviewer for this excellent suggestion, following paragraph is included in the revised manuscript.
Cerebellin-1 (Cbln1) is an ECM glycoprotein involved in cell adhesion that may play a role in cerebellar ataxia. Cbln1 is released from the parallel fibers of cerebellar granule cells and has been shown to play a role in the synaptic organization of the cerebellum [57]. In particular, Cbln1 can bind neurexins to the amino-terminal domains of GluD2 (glutamate receptor family member delta-2) on Purkinje cells, promoting synapse creation between parallel fibers and Purkinje cells [58]. Cbln1-null mice display motor deficits similar to that of cerebellar ataxia and Suzuki et al. demonstrated that these motor deficits could be ameliorated in adult mice using a synthetic synaptic organizer protein deemed CPTX [59]. CPTX utilizes the modular architecture of Cbln1 and a neuronal pentraxin (NP1) in an attempt to form excitatory synapses by physically bridging pre- and postsynaptic sites. After injecting CPTX into cerebellar lobules VI and VII of adult Cbln1-null mice, they found that CPTX treatment partially restored the synapses between parallel fibers and Purkinje cells and improved motor deficits of injected Cbln1-null mice. This effect decayed after the initial injection which suggests that continued presence of CPTX is necessary for synaptic maintenance.
These studies indicate that disruptions in ECM maintenance may underlie the dysfunction in the SCAs. Regulation of the ECM contributes to neuronal capabilities since it provides the scaffold for various structures critical for forming and maintaining effective connections between neurons. These include neuronal outgrowth and maintenance as well as ECM components involved in synaptic plasticity, formation, and regulation [29]. Prolonged disruption of these pathways could lead to disruptions in proper synapse formation and transmission. Synaptic pathology within the CNS has been noted in SCA1 mouse models, one of the most prominent being the altered climbing fiber innervation seen in cerebellar Purkinje cells [60]. This synaptic alteration is accompanied by notable dendritic atrophy as the disease progresses. Decreased expression of Cbln1 and dysregulation of other ECM genes may contribute to the loss of synapses and dendritic atrophy in SCA1 cerebella.
- In several places the authors refer to some changes in MMPs and claim that this would result in ECM changes. This is too superficial. They need to introduce major MMPs and other extracellular proteases digesting BM or PNN components in the introduction and then refer to changes in these specific proteases and their possible effects on specific ECM forms and functions.
We thank the reviewer for this excellent suggestion, and in response following paragraphs are included in revised manuscript:
In the introduction:
The components of ECM that regulate its degradation are enzymes including matrix metalloproteinases (MMPs), disintegrin and metalloproteinase with thrombospondin type 1 motif (ADAMTS), hyaluronidases/chondroitinases, and plasmin, and their regulators including tissue inhibitors of metalloproteinases (TIMPs), tissue plasminogen activator (tPA), and plasminogen activator inhibitor. MMPs are a family of zinc-dependent endoproteases comprising 28 individual members involved in several physiological processes, including tissue morphogenesis and cell migration, and pathophysiological processes such as inflammation and cancer [20]. Activity of MMPs is regulated by their expression, processing, and expression of their regulators. MMPs promote the turnover of collagens, elastin, gelatin and other matrix glycoproteins and proteoglycans that compose the ECM. Additionally, MMPs directly degrade ECM proteins such as aggrecan, laminin, entactin and fibronectin [21]. Importantly for PNNs, gelatinases such as MMP-2 and MMP-9 degrade aggrecan, laminin and fibronectin [8]. MMPs can be released by a variety of cell types, including microglia [22]. Degradation of PNNs by MMPs in disease can contribute to PV interneuron dysfunction, and as a consequence alter the balance of excitation and inhibition in the affected brain regions. Loss of PNNs can also lead to reduced levels of growth factors, loss of neuroprotection against oxidative stress, altered concentration of ions in neuronal microenvironment, as well as perturbed expression of NT receptors and ion channels causing maladaptive neuroplasticity [23][24][25].
In 2.1.:
These include Adamts1, encoding ADAMTS1 protein whose major substrates are lecticans, but that can also contribute to cell polarization and migration via regulation of cellular Rho-GTPases activities [33][34][35]. Cathepsin H (Ctsh) encodes for serine protease that was shown to increase expression of MMP genes through degradation of histone deacetylases (HDACs)[36]. Interestingly Plat and Plaur, two genes involved in plasmin activation, are regulated in the opposite directions. Plasminogen activator (Plat) encodes for a protein that catalyzes the plasminogen-plasmin conversion to modulate laminin degradation and activation of trophic factors such as nerve growth factors as well as activation of microglia. Plaur encodes for the receptor urokinase plasminogen (u-PAR), which upon binding to the protease urokinase-type plasminogen activator (uPA or urokinase) intitaties plasmin formation and promotes localized ECM degradation. As Plaur is expressed at higher levels and Plat at lower levels in the IB4+ DRG neurons, this suggests that fine tuning of ECM degradation towards a “golden middle” with neither too little nor too much of ECM remodeling could be essential for optimal neurite growth. In addition, a recent study indicated that mechanosensation of the stiffness of CNS tissue influences neurite outgrowth [37]. As the ECM contributes to overall CNS tissue stiffness, this further indicates the importance of balancing ECM deposition and degradation in brain physiology.
- There was a recent review on “A glial perspective on the extracellular matrix and perineuronal net remodeling in the central nervous system”. Thus, the discussion of microglia – ECM interaction appeared not an original issue. It also lacks in-depth discussion of downstream events after ECM remodeling by microglia.
We thank the reviewer for this comment. Our goal is to provide a brief overview. We have included references to the recent great reviews on the role of glia in ECM and expanded our discussion to address reviewer’s concerns.
As noted in previous sections, many cell types in the brain can express ECM genes. Due to their reactive changes with aging and in disease, and active role in phagocytosis, here we will focus on a few studies investigating the role of microglia in regulating the balance of ECM degradation and deposition, and in particular brain specific ECM component perineuronal nets. These studies provide insight into how microglia regulate PNNs and how microglial dysregulation of PNNs may contribute to neurodegenerative disease. We recommend other great reviews for a more in-depth overview of microglia and ECM[102][8][103].
Microglia play a role in the homeostasis of PNNs, as was demonstrated in a recent study by Nguyen et al. [106]. The authors show that enriched environment promotes IL-33 production by hippocampal neurons, promotes microglia-dependent ECM remodeling, hippocampal dendritic spine formation, synapse plasticity, and is required for memory consolidation. IL-33 induces changes in microglial gene expression including Gas7, a gene that plays a role in phagocytosis, and Marco, a gene that plays a role in debris clearance. This indicates that IL-33 can induce ECM phagocytosis by microglia. Moreover, superresolution imaging and 3D reconstruction revealed that IL-33 KO mice exhibited reduced levels of the PNN component aggrecan within microglial lysosomes indicating that IL-33 promotes microglial phagocytosis of PNN during learning. In addition, IL-33 induced the expression of genes involved in ECM degradation including Mmp14, Mmp25, and Adamts4, indicating that microglia also promote ECM degradation by secreting these enzymes. Importantly, this IL-33-induced and microglia-mediated remodeling of ECM was required for long term contextual fear memory. Intriguingly, microglia, ECM clearance, and IL-33 are all implicated in aging and mouse models of AD, suggesting that dysregulation of ECM clearance by microglia may contribute to age and AD cognitive decline [106].
Another study by Strackeljan et al. investigated the effects of reversible pharmacological microglial depletion on remodeling of PNNs in the CA1 region of the hippocampus, a region predominantly involved in learning [107]. Microglia depletion with PLX, an inhibitor of colony-stimulating factor 1 receptor (CSF1R) signaling, decreased the size of PNN holes and elevated the expression of the surrounding ECM. Furthermore, microglia depletion increased the expression of perisynaptic ECM component brevican and the number of excitatory presynaptic puncta labeled with vesicular glutamate transporter 1 (VGLUT1). This provides further evidence that in healthy brains microglia regulate PNNs and could potentially alter the physiology of PNN positive neurons in regions associated with memory.

Reviewer 2 Report
The authors intended to critically review recent findings regarding the relevance of ECM in brain development and in the alterations of ECM-associated genes and gene expression pathways in different neurological disorders, such as epilepsy, neuropathic pain, spinocerebellar ataxia type 1, etc. They also discuss the role of transcription factor hypoxia-inducible factor 1 (HIF-1) on ECM and genes involved in ECM remodeling and microglia in regulating perineuronal nets (PNNs).
There is a lot of space for improvements in the review.
The review could be better organized (order of the paragraphs and paragraphs' placed under adequate subtitles).
Preciseness and accuracy in citing original articles are needed.
For example, instead of citing article numbered (1) Milosevic NJ et al. 2014 in text line 32, there should be awarded paper by Sykova and Nicholson 2008. The article by Milosevic et. a. 2014. fits in the paragraph related to brain development (text lines 55-61, article text line 56) and the section about developmental disorders associated with ECM alterations (paragraph 71-81, text line 74).
The only figure, Figure 1, although very beautiful, is not informative, self-explanatory, and not described adequately or fully. It is tough to understand what is the purpose of this figure.
Instead of complex drawings, a schematic illustration of the message the authors wanted to convey will be of more value to the audience.
Author Response
We thank the reviewers for their helpful comments and suggestions that have improved our manuscript.
Our point to point responses (in blue) to reviewer’s comments (in italics) are bellow.
Reviewer 2
- The review could be better organized (order of the paragraphs and paragraphs’ placed under adequate subtitles).
We thank the reviewer for this suggestion. We have reorganized the review, added subtitles, and did our best to increase the flow within and between paragraphs.
- Preciseness and accuracy in citing original articles are needed. For example, instead of citing article numbered (1) Milosevic NJ et al. 2014 in text line 32, there should be awarded paper by Sykova and Nicholson 2008. The article by Milosevic et. a. 2014. fits in the paragraph related to brain development (text lines 55-61, article text line 56) and the section about developmental disorders associated with ECM alterations (paragraph 71-81, text line 74).
We have corrected this reference and did our best to include original citations throughout the review.
- The only figure, Figure 1, although very beautiful, is not informative, self-explanatory, and not described adequately or fully. It is tough to understand what is the purpose of this figure. Instead of complex drawings, a schematic illustration of the message the authors wanted to convey will be of more value to the audience.
We have revised and simplified Figure 1.

Reviewer 3 Report
Review of the manuscript entitled: Extracellular matrix regulation in physiology and in brain disease. The manuscript is interesting but some corrections should be made. In abstract, clear aim of the manuscript should be added e.g. "The aim of the present study was to ...". I believe that the manuscript's introduction is appropriate and legibly written. Unfortunately, the authors did not mention the extremely important role of elastin derived peptides (EDPs) in the nervous system.
Line 36 now is “elastins and collagen” should be “elastin and collagens” there is one elastin and many types of collagen in the body ;)
I believe that EDPs should be mentioned in chapter two because they have proven properties in inducing neurodegenerative diseases.
Between the lines 120-155 there is no reference at all, why? Similar on page 5 and 6, only 3 references. Since this is a review article, authors should not limit references.
Line 74 “metalloproteases (MMPs)” the correct abbreviation is “matrix metalloproteinases”
Maybe in chapter 3, it would be appropriate to mention the aryl hydrocarbon receptor, because it is extremely important in interactions with HIF-alpha.
Line 283 “AD” abbreviation should be used.
Line 289 reference should be corrected.
Author Response
We thank the reviewers for their helpful comments and suggestions that have improved our manuscript.
Our point to point responses (in blue) to reviewer’s comments (in italics) are bellow.
Reviewer 3
- Unfortunately, the authors did not mention the extremely important role of elastin derived peptides (EDPs) in the nervous system.
We thank the reviewer for this excellent suggestion. Following paragraph discussing the role of EDPs is included in the revised manuscript.
Aging is an important risk factor in most neurodegenerative diseases. One way by which aging can contribute to neurodegenerative diseases is via age-induced ECM degradation. In addition to causing an imbalance between ECM degradation and deposition, degradation of the ECM produces signaling peptides, such as elastin derived peptides (EDPs)[61]. Elastin is an important component of the ECM, in particular in vascular BM where it mediates elasticity of arteries. Elastin has extraordinary stability and an extremely low turnover rate; radiocarbon prevalence data indicates that elastin in the lungs is the same age as human subjects [62]. This indicates that elastin degradation is potentially irreversible or irreparable in aging and neurodegenerative or pathophysiological conditions [63]. While being generally resistant to proteolysis, elastin can be degraded by proteinases such as MMPs. Moreover, when elastin degrades, newer peptides form called elastin-derived peptides (EDPs) and the levels of EDPs increase with aging [64]. While still largely understudied, the increased presence of EDPs with age supports the idea that they play an important role in the progression and development of age-related disease disorders [65]. Importantly, when EDP binds to receptor elastin binding protein (EBP), it activates ERK and AKT kinases. This can result in increased expression of Mmps, which can then further degrade the ECM and promote inflammation [66]. In addition, elastin-like polypeptides (ELPs) can induce expression of genes such as Beta secretase1 (BACE1) that can directly contribute to the deposition of amyloid beta plaques–a hallmark of AD [67]. The processing of amyloid precursor protein (APP) into amyloid beta (Aβ) leads to the formation of Aβ plaques in the brain. Ma et al. showed that ELPs can induce overproduction of Aβ in a Chinese hamster ovary cell line that stably expresses mutant human amyloid precursor protein (7PA2 cells). They also found that ELP injection in wild-type C57BL/6 mice caused a significant upregulation of Aβ production. Treatment with ELPs in mice showed more than just upregulation of Aβ levels; these mice exhibited both pathological and neurobehavioral AD phenotypes confirming a relationship between ECM degradation and pathogenesis of AD. Together these studies suggest that elastin in brain ECM may be an important factor in initiating the neurodegenerative deficits and pathological changes due to AD progression.
- Line 36 now is “elastins and collagen” should be “elastin and collagens” there is one elastin and many types of collagen in the body.
We apologize for this mistake that is now corrected.
- Between the lines 120-155 there is no reference at all, why? Similar on page 5 and 6, only 3 references. Since this is a review article, authors should not limit references.
We thank the reviewer for pointing this, we have increased references from 52 to 115.
- Line 74 “metalloproteases (MMPs)” the correct abbreviation is “matrix metalloproteinases”
We apologize for this mistake, it is corrected.
- Maybe in chapter 3, it would be appropriate to mention the aryl hydrocarbon receptor, because it is extremely important in interactions with HIF-alpha.
We thank the reviewer for this excellent suggestions. Following discussion of AhR is included in revised manuscript.
Like HIF-1, Aryl hydrocarbon receptor (AhR) is a transcription factor implicated in aging, neurodegenerative disease, and ECM regulation. AhR and HIF-1 are both members of the basic helix-loop-helix-PER-ARNT-SIM (bHLH-PAS) protein family. These pathways have a complex relationship such that their simultaneous activation can have enhancing or antagonistic effects on downstream gene targets depending on the specific cell type and type of exposure [92][93]. AhR activity could also indirectly increase HIF-1 activity by decreasing the expression of PHD to disinhibit the hypoxia dependent α subunit [94]. AhR is activated by xenobiotic compounds such as dioxins that are released into the environment by industrial activities. AhR ligand binding causes a conformational change that exposes a nuclear localization signal which then allows for AhR translocation into the nucleus. Upon nuclear entry AhR dimerizes with HIF-1β, also known as Aryl hydrocarbon receptor nuclear translocator (ARNT), and interacts with xenobiotic response elements to modulate gene expression. For example, 2,3,7,8-Tetrachlorodibenzo-p-dioxin (TCDD) is an environmental chemical that can bind AhR, allowing it to enter the nucleus and stimulate the expression of metabolizing genes like cytochrome p450 1 family members [95]. In the neurodegenerative diseases PD and HD, AhR may contribute to pathogenesis by altering the expression of ECM genes including MMPs, TIMPS, genes involved in cell adhesion, cell-cell or cell-matrix interactions, as well as by promoting neuroinflammation [96][94]. In a mouse model of liver fibrosis TCDD AhR activation increased the expression of MMP-3, MMP-8, MMP-9, MMP-13, TIMP-1, plasminogen activator genes, and procollagen genes [97]. In human orbital fibroblasts AhR activation by the AhR ligand 6-formylindolo[3,2b]carbazole (FICZ) led to the expression of MMP-1 in an AhR and ARNT dependent manner and decreased collagen levels in a fibrosis model [98]. This evidence supports a role for AhR in ECM regulation, but further research is required to understand how AhR signaling alters ECM in the brain and how ECM gene regulation is impacted by simultaneous AhR and HIF-1α activity.
Altogether, this suggests that hypoxia can promote ECM dysregulation directly through its transcriptional activity on ECM components and indirectly in a mechanism that involves inflammation which promotes glial release of ECM modifying proteases like MMPs. The increased expression of MMPs could contribute to pathology and explain selective cell death in neurodegenerative disease. In a mouse model of ALS, reducing MMP-9 levels ameliorated neuronal loss and the expression of inflammatory proteins while extending survival [99]. Moreover, a transcriptomic study that assessed DEGs between spared and vulnerable motor neuron populations in a mouse model of ALS found MMP-9 was strongly expressed in vulnerable but not spared motor neurons and that reducing MMP-9 levels improved pathology [100]. Hypoxia, neuroinflammation and ECM dysregulation are all common themes in neurodegenerative disease, and it is possible that HIF-1α transcriptional activity could be a common mediator of the changes in the ECM in these conditions. To understand how hypoxia and HIF-1 alters the ECM it will be important to understand the crosstalk between HIF-1 and AhR. Moreover, another level of complexity arises from the fact that HIF-1β may also be regulated by hypoxia [101]. Additional investigation on if HIF-1β expression is altered by hypoxia in neural cell types and how mechanistically this occurs is necessary to understand the extent to which HIF-1 activity may alter the ECM. Further research is required to understand the cell types involved in hypoxia induced ECM regulation, how ECM dysregulation may contribute to disease, and how HIF-1 could be a therapeutic target to ameliorate changes in the ECM.

Round 2
Reviewer 1 Report
The authors addressed my comments and improved the review. It contains some novel and stimulating points.
Author Response
We thank the reviewer for time and helpful suggestions!
Reviewer 2 Report
Please cite the following recent papers related to ECM in epilepsy and hypoxia, where appropriate:
Sitaš et al.Reorganization of the brain extracellular matrix in hippocampal sclerosis // International journal of molecular sciences, 23 (2022), 15; 8197, 19 doi:10.3390/ijms23158197 Trnski et. al.The signature of moderate perinatal hypoxia on cortical organization and behavior: altered PNN-parvalbumin interneuron connectivity of the cingulate circuitries // Frontiers in cell and developmental biology, 10 (2022), 810980, 17 doi:10.3389/fcell.2022.810980Author Response
Response to reviewer 2
We thank the reviewer for this suggestion.
- Please cite the following recent papers related to ECM in epilepsy and hypoxia, where appropriate:
Sitaš et al.Reorganization of the brain extracellular matrix in hippocampal sclerosis // International journal of molecular sciences, 23 (2022), 15; 8197, 19 doi:10.3390/ijms23158197 Trnski et. al.The signature of moderate perinatal hypoxia on cortical organization and behavior: altered PNN-parvalbumin interneuron connectivity of the cingulate circuitries // Frontiers in cell and developmental biology, 10 (2022), 810980, 17 doi:10.3389/fcell.2022.810980.
Based on the reviewer comment we included following sentences describing and citing the above references in the revised manuscript.
In epilepsy paragraph:
Recent study demonstrated reduction in PNNs and significantly alltered expression pattern of versican, neurocan, aggrecan, and WFA-specific glycosylation in the hippocampus of patients with drug-resistant mesial temporal lobe epilepsy (MTLE)[46].
In hypoxia:
Even a single hypoxic event can induce long lasting changes in the ECM as demonstrated in perintal rats which exhibited altered PNN composition in the cingulate cortex that persisted into adulthood from exposure to a single hypoxic event [80]
In references:
- Sitaš B, Bobić-Rasonja M, Mrak G, Trnski S, Krbot Skorić M, Orešković D, et al. Reorganization of the Brain Extracellular Matrix in Hippocampal Sclerosis. Int. J. Mol. Sci. 2022;23.
- Trnski S, Nikolić B, Ilic K, Drlje M, Bobic-Rasonja M, Darmopil S, et al. The Signature of Moderate Perinatal Hypoxia on Cortical Organization and Behavior: Altered PNN-Parvalbumin Interneuron Connectivity of the Cingulate Circuitries. Front. Cell Dev. Biol. 2022;10:1–17.
.
